# A deep learning model for predicting next-generation sequencing depth from DNA sequence

Jinny X. Zhang[1,2,6], Boyan Yordanov [3,4,6], Alexander Gaunt[3,6], Michael X. Wang [1,6], Peng Dai[1], Yuan-Jyue Chen [5], Kerou Zhang[1], John Z. Fang [1], Neil Dalchau [3], Jiaming Li [1,2], Andrew Phillips [3✉] & David Yu Zhang[1,2✉]

Targeted high-throughput DNA sequencing is a primary approach for genomics and molecular diagnostics, and more recently as a readout for DNA information storage. Oligonucleotide probes used to enrich gene loci of interest have different hybridization kinetics, resulting in non-uniform coverage that increases sequencing costs and decreases sequencing sensitivities. Here, we present a deep learning model (DLM) for predicting Next-Generation Sequencing (NGS) depth from DNA probe sequences. Our DLM includes a bidirectional recurrent neural network that takes as input both DNA nucleotide identities as well as the calculated probability of the nucleotide being unpaired. We apply our DLM to three different NGS panels: a 39,145-plex panel for human single nucleotide polymorphisms (SNP), a 2000-plex panel for human long non-coding RNA (lncRNA), and a 7373-plex panel targeting non-human sequences for DNA information storage. In cross-validation, our DLM predicts sequencing depth to within a factor of 3 with 93% accuracy for the SNP panel, and 99% accuracy for the non-human panel. In independent testing, the DLM predicts the lncRNA panel with 89% accuracy when trained on the SNP panel. The same model is also effective at predicting the measured single-plex kinetic rate constants of DNA hybridization and strand displacement.

[1] Department of Bioengineering, Rice University, Houston, TX, USA. [2] Systems, Synthetic, and Physical Biology, Rice University, Houston, TX, USA. [3] Microsoft Research, Cambridge, UK. [4] Scientific Technologies, London, UK. [5] Microsoft Research, Seattle, WA, USA. [6]These authors contributed equally: Jinny X. Zhang, Boyan Yordanov, Alexander Gaunt, Michael X. Wang. ✉email: andrew.phillips@microsoft.com; dyz1@rice.edu

With more than 3 billion DNA nucleotides in the haploid human genome, deep sequencing of the entire human genome for clinical applications is not economically feasible. Instead, researchers and diagnostic laboratories typically use targeted sequencing, in which a set of DNA hybridization probes is designed to bind and enrich the DNA regions of interest[1,2]. However, the DNA oligonucleotide probes in a targeted sequencing panel typically all have different kinetics and thermodynamics of binding to their respective targets. Consequently, a naively designed and synthesized panel of DNA probes will result in grossly different enrichment efficiencies for different genetic loci.

The sensitivity of NGS to a locus is directly proportional to the number of NGS reads that contain the locus (the locus's *sequencing depth*). Nonuniformity of sequencing depth either reduces the sensitivity at low-depth loci or necessitates additional sequencing to guarantee that all loci are sequenced to a minimum desired depth. Empirical optimization of an NGS panel's probe sequences and concentrations is time-consuming and labor-consuming, but currently cannot be avoided. A computational method to predict the sequencing depth based on probe sequence could inform the selection of probe sets with higher uniformity and modulation of probe concentrations to achieve higher uniformity.

The DNA biochemistry and biophysics literature contains several well-validated models of DNA structure, thermodynamics[3,4], and kinetics[5–8]. To model sequencing depth against probe sequences, ignoring our extensive knowledge of DNA biophysics and relying only on DNA sequence information would likely lead to suboptimal model performance. Simultaneously, we want to avoid extensive feature construction and curation, as such expert systems are generally labor-intensive to build and exhibit low generalizability to adjacent problems. Consequently, we decided to take a middle ground where we utilized only a small number of global (oligonucleotide molecule-level) features and local (individual nucleotide-level) features that can be fully autonomously computed by the well-accepted DNA folding software Nupack[9] (Fig. 1b).

Here, we constructed a deep learning model (DLM) for predicting NGS sequencing depth for a given oligonucleotide probe and characterized its performance on predicting the sequencing depths of three NGS panels, one with 39,145 probes against human single nucleotide polymorphisms (abbreviated as SNP panel), one with 2000 probes against human long non-coding RNA (abbreviated as lncRNA panel), and one with 7373 probes against artificially designed synthetic sequences for information storage (abbreviated as synthetic panel)[10]. The lncRNA panel serves as an independent test set for the SNP panel as its probes were separately designed and experimentally tested using the same library preparation method (Fig. 1a). Our DLM is based on a recurrent neural network (RNN) architecture to better capture both short-range and long-range interactions within the DNA probe sequence that can impact capture efficiency and speed.

## Results

**Design of the deep learning model.** In the genomics field, DNA probe oligonucleotide lengths range from 50–150 nucleotides (nt). Thus, in designing our DLM, we considered that the model should be generalizable to DNA sequences of different lengths. To this end, neural networks (NN) with a fixed number of input nodes, including conventional feed-forward NNs and convolutional NNs for image recognition[11], are not well-suited for DNA sequence inputs. Furthermore, from DNA thermodynamics and structure studies[3,9,12,13], we know that distal DNA nucleotides can hybridize to each other in secondary structures. These long-range interactions in DNA molecules are better captured by recurrent neural networks (RNNs), which have been applied commercially in speech recognition and natural language processing[14].

In brief, RNNs contain a number of internal hidden nodes, which are updated serially based on the ordered inputs and their current state values. RNNs have two primary implementations: long short-term memories (LSTMs) and gated recurrent units (GRUs). We chose to implement our DLM using GRUs because they have been reported to achieve similar performance using fewer computational resources[15]. Our DLM includes a total of four GRUs grouped into two sets (Fig. 1c): two GRUs for target sequence $T$, and two GRUs for probe sequence $P$. Although the target sequence is always the reverse complement of the probe sequence in our DLM model, we included separate GRUs for $T$ and $P$ both to ease the training of the model and to enable the DLM to be more generalizable to problems with asymmetric information on $T$ and $P$, such as in the strand displacement kinetics that we discuss later.

Each of the two GRUs for each oligonucleotide ($T$ or $P$) takes the sequence either in the direction from 5′ to 3′, or from 3′ to 5′ (Fig. 1c). Unlike biological polymerization reactions which have a clear 5′ to 3′ directionality, the hybridization process is equally likely to initiate on either end. For RNNs and GRUs, the last inputs tend to have a larger influence on the final state values of the hidden nodes, so the design decision to include sequences in both directions is aimed at reducing input direction bias.

For each GRU, at every single nucleotide there are three input variables: (1) a binary bit indicating whether the nucleotide is a purine (A or G), (2) a binary bit indicating whether the nucleotide is "strong" (G or C), and (3) an analog Nupack-computed probability $p_{unpaired}$ that the nucleotide is unpaired at the reaction conditions[9]. We chose to encode the identity of each nucleotide in two dimensions rather than a single dimension (e.g., $A = 1$, $T = 2$, $C = 3$, $G = 4$), in order to reflect the pairwise "distances" between any two nucleotides, based on DNA biochemistry knowledge. The unpaired probability of each nucleotide reflects our biophysical understanding that only unpaired nucleotides can participate in hybridization reactions; a paired nucleotide must first dissociate in order to allow new Watson-Crick base-pairing. $p_{unpaired}$ is calculated using Nupack and considers the ensemble of all possible secondary structures that can be adopted by each DNA molecule, rather than just the minimum free energy structure.

Each GRU was designed to have 128 hidden nodes ($\mathbf{h_t}$). All node values are initialized to 0 and updated based on each nucleotide's information. The hidden nodes of the RNNs represented potential patterns in the DNA sequence that the GRU could identify, and the final values of states after updating all nucleotides in the $T$ and $P$ sequences correspond to the presence or absence of those patterns. Thus, the number of hidden nodes in the RNN (currently 128) limited the maximum number of patterns that could be observed by the RNN. Preliminary studies showed similar prediction performance for GRUs with 128 internal states as for 256 internal states (data not shown), suggesting that 128 states were sufficient to capture the bulk of the patterns. Through the course of DLM training and weight updating through back-propagation, the GRU parameter weights were modified until they represented frequently observed patterns in the training data.

Downstream of the GRU, we used a conventional feed-forward neural network (FFNN) that takes as input the final state values of the hidden nodes of the GRUs (128 from $\mathbf{H}^{5'->3'}$ and 128 from $\mathbf{H}^{3'->5'}$). In addition to the hidden node values, the FFNN also takes as input 4 global features: the reaction temperature, the

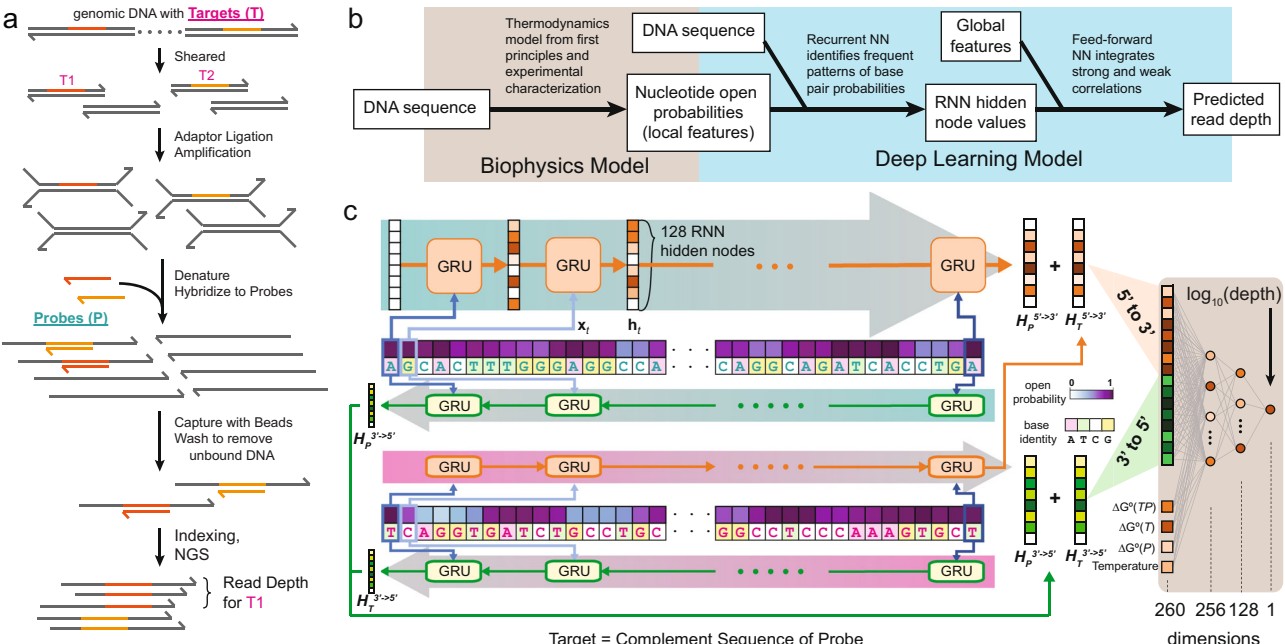

**Fig. 1 Overview of targeted NGS panel workflow and deep learning model (DLM) for predicting sequencing depth. a** DNA probes (*P*) are designed for hybridizing to target sequences (*T*) for subsequent solid-phase separation and enrichment. In NGS, each read corresponds to a randomly sampled DNA molecule from the enriched library, and NGS reads are bioinformatically aligned to the probe sequences using standard algorithms and software. See Supplementary Note 1 for further details on the NGS experimental and bioinformatic workflow. **b** Overview of NGS read depth prediction method. **c** The DLM consists of 4 recurrent neural networks implemented as gated recurrent units (GRUs) and 1 feed-forward neural network (FFNN). Each GRU has 128 internal state nodes. The final GRU node values for the target DNA sequence *T* and for the probe DNA sequence *P* from 5′ to 3′ ($H_T^{5'->3'}$ and $H_P^{5'->3'}$) are summed; likewise $H_T^{3'->5'}$ and $H_P^{3'->5'}$. These two hidden state node sum vectors are then concatenated into a vector of 256 node values, serving as input to the FFNN. In addition, 4 global parameters also serve as input to the FFNN, bringing the total inputs to 260. The output was a single node corresponding to the log predicted read depth $\log_{10}(depth)$ for the DNA target sequence.

predicted standard free energy of folding of probe($\Delta G°(P)$) and Target($\Delta G°(T)$), and the predicted standard free energy of formation of the *TP* double-stranded DNA molecule ($\Delta G°(TP)$). These global features were intended to capture properties of the *T* + *P* reactions that were not easily revealed by the base pair probabilities. Thus, a total of 260 nodes were used as the FFNN input. The FFNN network contained 2 hidden layers with 256 and 128 nodes, respectively; these values were picked arbitrarily based on our experience, and overall prediction performance did not appear to be sensitive to the dimensionality of the FFNN hidden layers.

**Training and validation of the DLM on NGS read depth.** The SNP panel and the synthetic panel were used to independently train the DLM, and sequencing depths were predicted in cross-validation for each panel individually. The lncRNA panel was used as a separate test set for the SNP panel since these two panels share the same library preparation method. The reason for doing so is because each NGS library preparation method has a large number of different experimental variables (experimental workflow, sample type, hybridization temperatures, etc.) that we felt were beyond the scope of the DLM. From a practical point of view, we expect that most users would aim to selectively optimize probe sequence and concentration to improve uniformity within one NGS library preparation method, rather than across different methods. Nevertheless, we still tried training and predicting between the SNP panel and the synthetic panel and summarized the results in Supplementary Note 3. Probe sequences and observed read depth can be found in Supplementary Data 1–3. Each of our NGS datasets consists of two parts: features generated from probe sequences and read depth measured with one NGS

library. For the SNP panel and the synthetic panel, we randomly split the data into 20 classes, and predictions of each class (5% of the total dataset) were obtained by a DLM trained on the remaining 19 classes (95% of the total dataset), as shown in Fig. 2a. Thus, a total of 20 DLMs were used in the 20-fold cross-validation predictions for evaluating prediction accuracy. Within each training set, the global features and the $\log_{10}(Depth)$ were standardized to have a mean of 0 and a standard deviation of 1. The mean and standard deviation of each training set was used to standardize the global features of the corresponding validation set, and to rescale the model predictions to their original mean and standard deviation. There are roughly 300,000 weight parameters in the DLM (illustrated in Fig. 1c); these were preset via Xavier initialization (uniformly distributed weights with standard deviation depends on the number of parameters in a layer) in order to alleviate the vanishing gradient problem for deep NNs[16].

During training, we iteratively minimized the Loss using gradient descent with an Adam optimizer[17] to update the network weights. The Loss here is proportional to the mean squared error between the predicted and experimental log sequencing depth. To minimize overfitting, we implemented an additional dropout layer after each hidden layer of the FFNN, in which 20% of parameters are randomly selected and prevented from updating in each training iteration. The DLM was implemented using Tensorflow[18], and DLM hyper-parameters include GRU hidden nodes (128), FFNN hidden nodes (256 and 128), batch size (999), learning rate (0.0001), and node dropout fraction (20%). We tried roughly 50 sets of different hyper-parameter values, and the values listed above appear to yield the shortest training time and best predictive performance. Training stops at epoch 250 and 1000 for the SNP panel and the synthetic panel, respectively. For the SNP panel, the training time for each epoch is roughly 10 s while taking less than 3 gigabytes

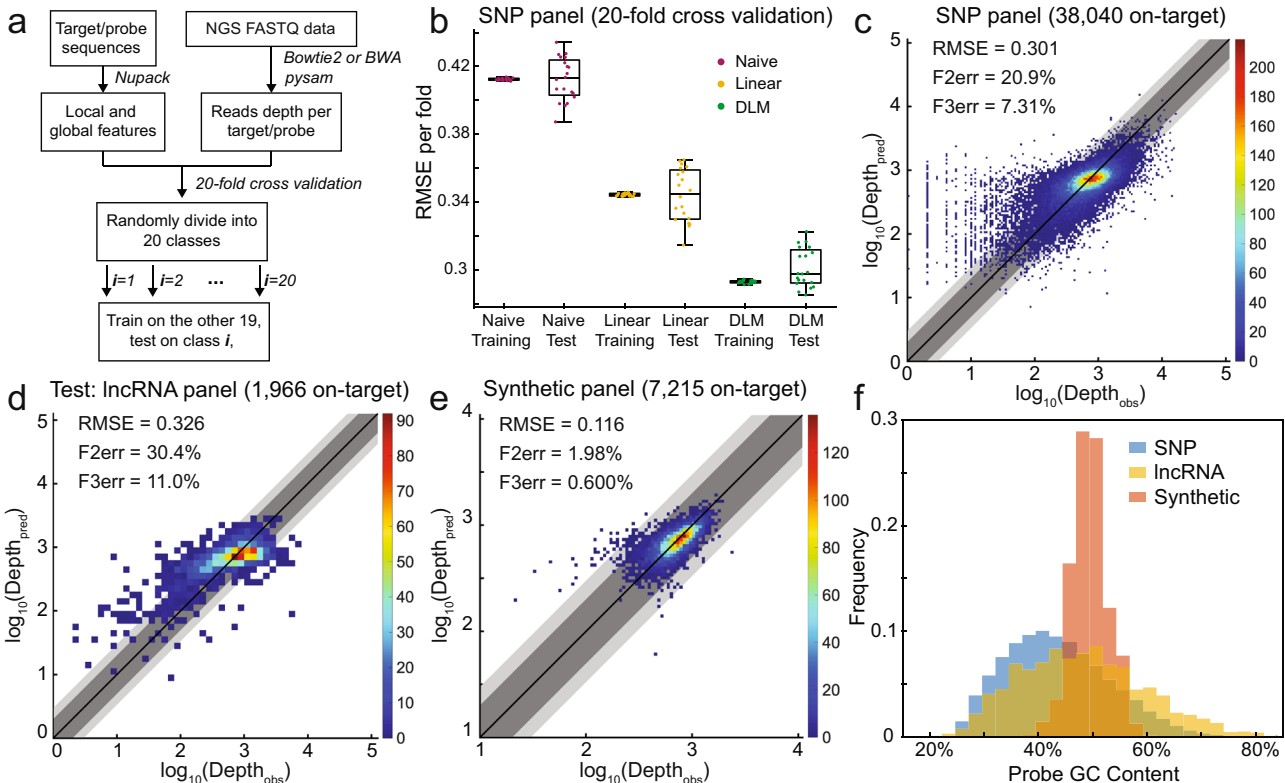

**Fig. 2 Cross-validation and independent test results of the DLM on predicting NGS depth. a** Data pre-processing and training pipeline. Raw FASTQ NGS reads were aligned to the target sequences[33,34], and the read depth of each template sequence was counted[35]. Separately, local and global features for each target/probe were computed using Nupack[9]. For the SNP panel and the synthetic panel, the complete dataset was split into 20 classes, and each class was predicted based on the DLM training results on the other 19 classes. **b** Summary of observed root-mean-square error (RMSE) for the SNP panel. Each point corresponds to the prediction results for one of the 20 validation classes; the naive model predicted $\log_{10}$(Depth) based on the mean of the training set, while the linear model fit $\log_{10}$(Depth) by the four global features (including the intercept). In box-whisker plots, the central mark indicates the median, and the bottom and top edges of the box indicate the 25th and 75th percentiles, respectively. The maximum whisker lengths are specified as maxima and minima. **c** Predicted vs. observed sequencing depth for the SNP panel. The plots are the aggregated results for all 20 validation classes. Dark gray shading marks the zones where the predicted and the observed read depth agreed to within a factor of 2; light gray shows agreement to within a factor of 3. F2err and F3err denote the fraction of sequences with depth predicted beyond a factor of 2 and 3, respectively. Colormap shows the number of probes within each 2-D bin. **d** Predicted vs. observed sequencing depth for the lncRNA panel, which contains a total of 2000 probes, where 34 probes with 0 depth were excluded. lncRNA panel was separately designed and experimentally tested using the same library preparation method as the SNP panel, serving as an independent test set. The DLM was first trained on the SNP panel with early stopping at 250 epoch and then predicted the lncRNA panel with the same model parameters. Read depth was scaled so that the average was the same as the SNP panel. **e** Predicted vs. observed sequencing depth for the synthetic panel, which contains a total of 7373 probes, where 158 probes with 0 depth were excluded. The plots are the aggregated results for all 20 validation classes. **f** Histogram of G/C content of the probe sequences from the three panels. The SNP panel and the lncRNA panel contain a greater variability in G/C content since the synthetic panel is procedurally designed and has a tighter G/C content distribution.

memory of a graphics processing unit, and feature generation using Nupack takes about 0.5 s per probe sequence on a conventional desktop computer.

**DLM performance and reproducibility on NGS datasets**. Figure 2b summarizes the root-mean-square error (RMSE) of the DLM predicted values of $\log_{10}$(Depth) based on different sequences vs. the actual observed NGS read depth for the SNP panel comprising 39,145 probes synthesized as a pool by Twist Biosciences. Notably, all the probes hybridize under the same temperature (65 °C) and bear the same length of 80 nt, excluding the two adapters (30 nt) at both ends for probe amplification. Of the 39,145 probes, NGS results showed 0 reads on 1105 probes. Our previous studies on Twist oligonucleotide pools suggest that the lack of sequencing reads for these may indicate difficulties with probe synthesis[19]. Consequently, we chose to exclude these probes with 0 observed NGS reads in order to eliminate the possibility of training against noise.

The DLM yielded an average RMSE of roughly 0.30 on both the training classes and the test classes. For comparison, a naive model of predicting sequencing depth only based on the mean $\log_{10}$(Depth) of all observed sequences produced an RMSE of 0.41. A linear regression model that fit the four global features (including the intercept) against the $\log_{10}$(Depth) produced an RMSE of 0.34 (Fig. 2b). Figure 2c plots the comparison of predicted and measured sequencing depths for each sequence, where the 20 validation sets are aggregated. From this figure, we see that a significant contributor to our DLM's RMSE is a subset of DNA sequences that are observed to have very low $\log_{10}$(Depth) (e.g., 0.3, corresponding to a depth of 2), but predicted to have $\log_{10}$(Depth) between 1 and 3.3. Further investigating the probe sequences, we found that most of the probes have low G/C content. Our interpretation of this phenomenon is that probes with lower expected read depth (e.g., probes with low G/C content) are more sensitive to random fluctuations (probe synthesis yield, hybridization yield, binding to plasticware, bridge PCR efficiency during Illumina NGS, etc.),

which would bias the observed $\log_{10}$(Depth). For example, suppose the expected read depth for a certain probe is 50 and the random fluctuation is ±45 with uniform distribution, then the observed depth ranges from 5 to 95 while the observed $\log_{10}$(Depth) ranges from 0.70 to 1.98. Note that the expected $\log_{10}$(Depth) is 1.70; thus there is a higher probability of $\log_{10}$(Depth) being lower than expected than being higher than expected (1.70 − 0.7 < 1.98 − 1.70). If the DLM predicts the expected $\log_{10}$(Depth), then there would be more probes whose observed $\log_{10}$(Depth) is lower than the predicted $\log_{10}$(Depth). However, the DLM cannot explain those random fluctuations solely based on probe sequences.

To validate our DLM in the practical scenario of optimizing new panels with a model trained on existing panels, we designed and tested the lncRNA panel comprising 2,000 DNA probes synthesized by Twist Biosciences. The lncRNA panel has the same library preparation method (experimental workflow, probe length, hybridization temperature, sample type, etc.) as the SNP panel, but differs in probe sequences, experimental operator, donor of DNA sample, sequencing instrument, and a batch of reagents. Figure 2d shows the predictions of lncRNA panel produced by a DLM trained on SNP panel with early-stop at epoch 250. Read depth of the lncRNA panel is scaled so that its average read depth is the same as the SNP panel. Despite the RMSE, F2err, and F3err of the lncRNA panel being slightly worse than the SNP panel (0.326 vs. 0.301, 30.4% vs. 20.9%, and 11.04% vs. 7.31%), the performance decrease may be attributed to experimental variations that are not related to the library preparation method. It is important to point out that the DLM was trained on read depth measured with only one NGS library of SNP panel, which greatly reduced the cost of training such a model. The results from the lncRNA panel indicate that the DLM is capable of generalizing different panels with the same library preparation method while being robust against experimental variations.

Figure 2e shows the DLM results applied to the synthetic panel comprising 7373 DNA probes against non-biological DNA sequences intended for DNA information storage applications. All the probes hybridize under 55 °C and share the same sequence length of 110 nt, excluding the two 20 nt adapters for probe amplification. As these sequences were procedurally generated to avoid known problematic DNA sequences, such as those with high or low G/C content (Fig. 2f), homopolymers, etc., there is much less variation in sequencing depth, to begin with. Nonetheless, the DLM is effective at predicting sequencing depths beyond the naive model and the linear regression model (Supplementary Note 3).

Our DLM in total contains over 300,000 parameters (e.g., node biases and node-node weights). A large number of parameters leads to potential concerns regarding overfitting and model reproducibility. To address this, we next performed 15 independent 20-fold cross-validation runs on the SNP panel, in order to characterize the reproducibility of the model (Fig. 3). For each cross-validation run, the whole dataset was randomly grouped into 20 classes and each of the 300 ($15 \times 20$) DLMs was initiated with different weight parameters. All 300 DLMs consistently reached early-stop at rough epoch 250 (Fig. 3a), and the predicted sequencing depths showed high pairwise concordance (Fig. 3b,c). Across the 105 pairwise comparisons of the 15 cross-validation runs, we observed a Pearson's r value of no less than 0.975. Consequently, we believe that our approach produces DLMs with fairly consistent predictions despite variations in parameter initialization and training sets.

**DLM prediction of single-plex DNA hybridization and strand displacement rate constants**. Based on our understanding of DNA and NGS, we believe that NGS read depth is primarily dependent on the yield and speed of DNA probe hybridization, and secondarily by chemistry-specific biases. Consequently, our DLM should also be effective at predicting the rate constants of hybridization of DNA (Fig. 4a). To further challenge our DLM and to highlight the effectiveness of our DLM approach, we further applied the DLM to the prediction of a related DNA mechanism, strand displacement[20,21] (Fig. 4b). Unlike NGS experiments in which thousands of DNA probes and targets are simultaneously hybridizing, for DNA hybridization and strand displacement rate constant prediction, we use time-based fluorescence data in which a single target and probe species are observed with high time and yield resolution (Fig. 4c, Supplementary Note 2). Sequences of targets, blockers, and probes can be found in Supplementary Data 4 and 5. See also ref. [8] for additional explanation on hybridization reaction kinetics experimental details.

Experimental hybridization rate constants are taken from ref. [8], and experimental strand displacement rate constant data are collected for this work. Supplementary Note 2 describes details of how best-fit rate constants are fitted to fluorescence-based kinetics data. Figure 4d shows the DLM prediction vs. experimental best-fit rate constants for 210 hybridization reactions and 211 strand displacement reactions. Note that the hybridization and strand displacement experiments used the same set of 100 probe sequences of 36 nt and each probe was tested under different temperatures (28 °C to 55 °C), producing a total of 421 data points. Data points sharing the same probe sequence are grouped as one class. Here, we performed 100-fold leave-one-class-out (LOCO) prediction rather than 20-fold cross-validation, because the smaller number of data points would lead to significant biases due to small sample sizes of the test set; see Supplementary Note 3 for details. The variation in rate constants observed is similar to the variation in NGS sequencing depths (4 logs), though the latter is possibly somewhat smaller due to the saturation of hybridization for the timescales of NGS hybrid-capture reactions. In addition, we compared the DLM predictions of hybridization rate constants with the predictions from a previous expert system machine learning approach based on weighted neighbor voting[8]. Figure 4e shows that their prediction results are similar.

The purpose of this sub-study was to see if the DLM could be used for non-NGS applications of nucleic acid molecular diagnostics, such as those based on qPCR[22,23] and electrochemistry[24]. Importantly, our DLM was trained simultaneously on both the hybridization and the strand displacement rate constant datasets. Because the target $T$ and probe $P$ sequences for the hybridization reactions are identical to that of strand displacement reactions, the difference between the two is manifested only in the predicted probability of each nucleotide being unpaired. This information alone was enough to communicate to the DLM the distinction between hybridization and strand displacement, and no special case handling or neural network architecture modification was needed to accommodate strand displacement.

**Contribution of different features to DLM performance**. The architecture of the DLM and the local and global features were initially decided based on our understanding of the behavior of DNA, rather than through knowledge-free exploration. Consequently, it is possible that some of the features are not directly relevant to NGS depth or hybridization/strand displacement rate constants. To test this hypothesis, we next constructed a series of DLMs in which different features were removed from the model (Fig. 5).

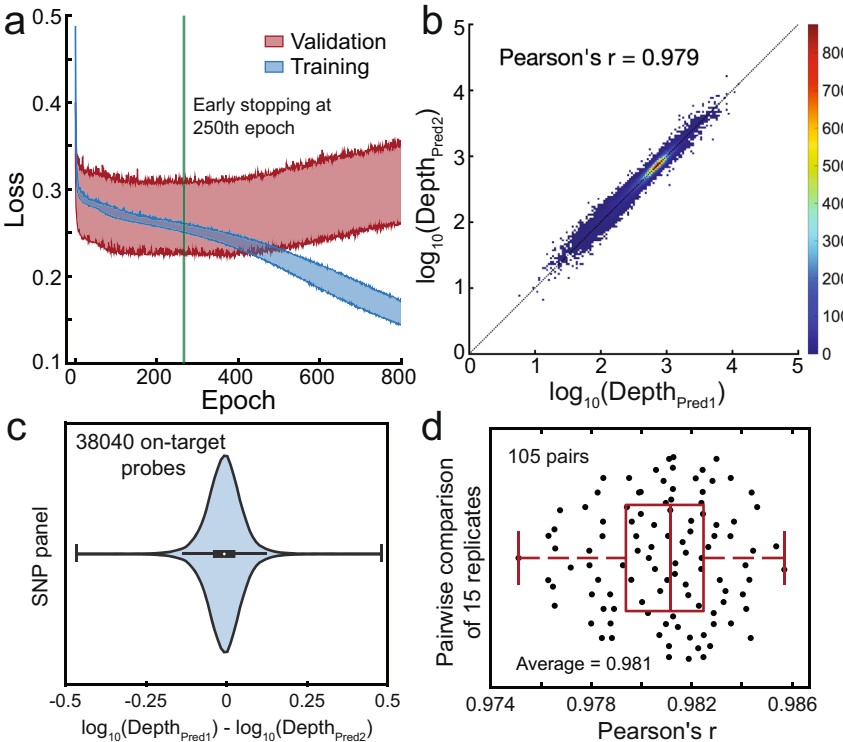

**Fig. 3 Reproducibility of DLM predictions for SNP panel. a** The plot of Loss vs. training epoch, showing 300 (15 × 20) DLMs from 15 independent 20-fold cross-validation runs. The shaded blue area shows the upper and lower limit of Loss of the 300 training sets, while the shaded red area shows the upper and lower limit of Loss of the 300 validation sets. At above 250 epochs, the training set Loss consistently continues to decrease but the validation set Loss starts increasing, indicating overfitting. Consequently, we halt DLM training at 250 epochs. **b** Comparison of read depth prediction results on the SNP panel from two independent cross-validation runs. For each cross-validation run, the whole dataset was randomly grouped into 20 classes and each of the 300 DLMs was initiated with different weight parameters. See Supplementary Note 3 for additional comparison results. **c** The deviation between predicted read depths from 2 independent cross-validation runs. In the box-whisker plot, the central mark indicates the median, and the bottom and top edges of the box indicate the 25th and 75th percentiles, respectively. The maximum whisker lengths are specified as 1.5 times the interquartile range. **d** Summary of pairwise comparison results for 15 independent cross-validation runs (105 total comparisons). In the box-whisker plot, the central mark indicates the median, and the bottom and top edges of the box indicate the 25th and 75th percentiles, respectively. The maximum whisker lengths are specified as maxima and minima.

We found that 3 of the 4 global features, individually, had essentially no impact on any of the predictions. Both sets of local features (sequences and base probabilities) were important for some aspects of the DLM prediction, but it appears that the two are interchangeable for predicting the NGS depths of the SNP panel. Examining the global features more closely, we note that temperature T is the same for all sequences within a panel, so it is tautological that the DLM cannot learn any impact from changing T.

The standard free energy of formation of the target-probe duplex $E_{TP}$ likely did not matter because the lengths of all probes/targets were long enough that the probe binding was no longer limited by its thermodynamics. Finally, the standard free energy of folding of the target by itself $E_T$ likely did not matter because it was not and could not be accurately calculated: whereas the probe P has a homogenous molecular population with a well-defined sequence, the target T is a heterogeneous mixture constructed through randomized physical fragmentation of human genomic DNA. Consequently, the 5′ and 3′ overhang sequences of the target are highly variable, and cannot be reflected as a single sequence. Prediction accuracies of all feature-reduced DLMs are summarized in Supplementary Note 4. For predicting DNA hybridization and strand displacement rate constants, the performance of even reduced models is in general much better than random guess models (Supplementary Note 5).

## Discussion

Targeted high-throughput sequencing of DNA has become a dominant method for biological and biomedical research, and furthermore is becoming standard of practice for cancer treatment[25]. More recently, targeted sequencing has been explored as a method for random-access readout of information stored densely and for the long term in DNA[10]. Although DNA sequencing costs are exponentially decreasing over the years[26], poor sequencing uniformity would waste a large majority of reads sequencing high-depth targets redundantly and providing insufficient information on low-depth targets. Consequently, a strong need exists to rationally design NGS panels with high uniformity.

However, predicting DNA hybridization kinetics and efficiency is extremely difficult even for experts in single-plex settings[8]. In a complex multi-component system that is hybrid-capture target enrichment, prediction of sequencing depth becomes intractable for first-principles biophysical models. At the same time, the large size of NGS datasets renders the problem well-suited for machine learning.

Deep learning leverages large datasets to autonomously discover weak correlative features between inputs and outputs. This has led to deep learning becoming dominant in computer vision and other areas with large available datasets. On the other hand, simpler statistical models (e.g., multivariate linear regression) remain dominant in the natural and biomedical sciences[27,28]

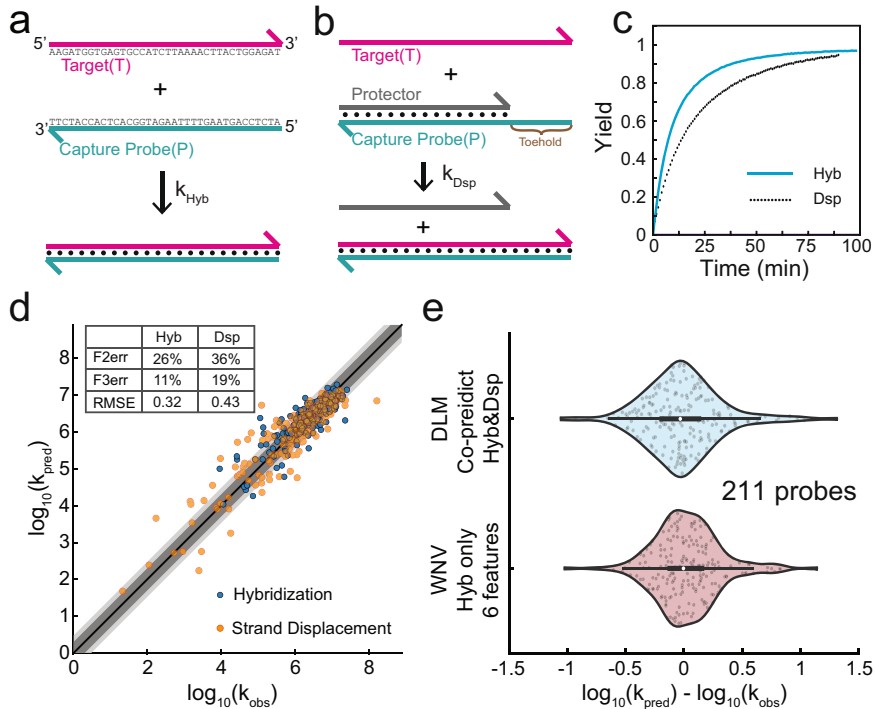

**Fig. 4 Applying our DLM to the prediction of single-plex hybridization and strand displacement rate constants.** Schematic of (**a**) hybridization and (**b**) strand displacement reactions. **c** Sample kinetics traces of hybridization and a strand displacement reaction. Reaction yields inferred through fluorescence; see Supplementary Note 2 for experimental and data processing details. **d** Accuracy of DLM predictions of hybridization (blue) and strand displacement (orange) rate constants. Dark gray shading marks the zones where the predicted and the observed read depth agreed to within a factor of 2; light gray shows agreement to within a factor of 3. Due to the small number of data points here (421), we implemented prediction based on 100-fold leave-one-class-out (LOCO) rather than 20-fold cross-validation, due to the large expected variation in small validation classes. Note that strand displacement and hybridization reaction parameters were co-trained using the same DLM, and predictions were likewise made using a single DLM. **e** Comparing DLM prediction performance to a previous expert system machine learning approach based on weighted neighbor voting[8]. In box-whisker plots, the central mark indicates the median, and the bottom and top edges of the box indicate the 25th and 75th percentiles, respectively. The maximum whisker lengths are specified as 1.5 times the interquartile range. Prediction results are similar.

| | (Pr) (S) $E_T E_P E_{TP}$ T | (●) (S) $E_T E_P E_{TP}$ T | (Pr) (S) $E_T E_P E_{TP}$ T | (Pr) (●) $E_T E_P E_{TP}$ T | (Pr) (S) ▮ ▮ ▮ T | (Pr) (S) $E_T E_P E_{TP}$ T | (Pr) (S) $E_T E_P$ ▮ T | (Pr) (S) $E_T E_P$ ▮ T | (Pr) (S) $E_T E_P$ ▮ T |
|---|---|---|---|---|---|---|---|---|---|
| NGS depth (Human) | 0.301 | 0.310 | 0.301 | 0.310 | 0.315 | 0.302 | 0.304 | 0.309 | 0.301 |
| NGS depth (Synthetic) | 0.12 | 0.15 | 0.12 | 0.16 | 0.12 | 0.12 | 0.12 | 0.11 | 0.12 |
| Hyb. rate constant | 0.32 | 0.47 | 0.45 | 0.30 | 0.38 | 0.32 | 0.36 | 0.32 | 0.32 |
| Displacement rate constant | 0.43 | 0.67 | 0.77 | 0.40 | 0.52 | 0.40 | 0.52 | 0.42 | 0.42 |

RMSE

(Pr) Local Feature - Base pair open probability $p_{unpaired}$
(S) Local Feature - Sequence Identity(A/T/C/G)
$E_T$ Global Feature - standard free energy of Target $\Delta G^0(T)$
$E_P$ Global Feature - standard free energy of Capture Probe $\Delta G^0(P)$
$E_{TP}$ Global Feature - standard free energy of TP complex $\Delta G^0(TP)$
T Global Feature - Reaction Temperature
○ ▢ Feature Used
● ▮ Feature NOT Used

**Fig. 5 Assessing the importance of different components of the DLM to prediction accuracy, measured by RMSE.** We tested different DLM that lack some features, in order to assess which ones meaningfully contribute to prediction performance. Highlighted in red are the applications in which a reduced DLM model performs significantly worse than our default DLM. Columns highlighted in blue indicate the reduced DLMs with essentially identical performance to our default DLM. While some features are redundant for the specific applications considered here, we note that most features improve performance in at least one application.

where well-curated data for specific problems are scarce. Expert system machine learning approaches based on extensive manual feature construction and curation take an in-between approach, using expert knowledge to guide the construction of narrowly optimized prediction software, but are generally poor at generalization to similar problems.

In our DLM, we restricted our inputs to a limited set of global and local features that can be automatically computed based on

DNA sequence, in order to avoid the trap of labor-intensive and problem-specific model construction. Given both the DNA thermodynamics model inaccuracy/incompleteness[4] and the fact that we could not feasibly consider the intermolecular interactions from all 3+ billion nucleotides of the human genome, we believe that the Nupack-predicted base-pair probability values likely have significant error. Future models for more accurately predicting base-pair accessibility in a highly complex and

heterogeneous solution could prove crucial to further improve the prediction accuracy of the DLM.

DNA sequences are rather unlike most other inputs for problems solved by deep learning networks. DNA molecules are known to have long-range interactions where distal DNA nucleotides bind to each other. Moreover, there are not orderly "grammar" rules such as in natural language processing[29] that can be readily discovered by neural networks. Distal and inter-molecular DNA binding is essentially the effect of chemistry and does not necessarily need to conform to human intuition. These all contribute to the difficulty of building neural networks that accurately predict DNA behavior based on the sequence.

Conversely, once a neural network architecture is established to "understand" DNA sequences, it could hold potential for a large range of other nucleic acid-based problems. As a research example, a large range of non-coding RNAs[30] have been discovered; being able to predict their structures can provide insights into their function. As a biomedical example, codon optimization problems[31] for synthetic biology, including de novo construction of RNA-based drugs[32]. The incorporation of generalizable domain knowledge within deep learning architectures will be a key enabler for predicting behaviors for nucleic acids, given the impact of their sequences on form and function and the exponential number of possible sequences of given lengths.

## Methods

In this paper, we performed DLM training and validation on two types of datasets: NGS sequencing depth and DNA interaction kinetics rate constants. We would like to introduce experimental methods individually on both systems.

**NGS studies**. We have two sets of NGS read depth data from two different starting materials: sheared human genomic DNA fragments and synthetic DNA sequences. We start with the human genomic DNA panel experiment.

### Human genomic DNA NGS library preparation

*Target pool preparation from Human Genomic DNA*. Genomic DNA used in this NGS experiment was extracted from the buffy coat of a patient blood sample, using QIAamp DNA Blood Mini Kit (Qiagen). All extracted Genomic DNA was sheared for 5 min using the Covaris M220 Focused-Ultrasonicator and Holder XTU Insert microTUBE 130 L to generate products with a basepair-peak of 170–200 bp. This sheared DNA was quantified using Qubit 3.0 Fluorometer (Thermo Fisher Scientific) and Qubit dsDNA BR Assay Kit (Thermo Fisher Scientific). As demonstrated in main text Fig. 1. a, 25 ng of sheared genomic DNA was used for downstream end-prep ligation and size-selection using NEBNext Ultra II DNA Library Prep Kit for Illumina. The end product was quantified using qPCR and then sufficiently amplified with index primers. PCR amplicons were purified and size selected using Dual-Side size selection of SPRISelect Beads(Beckman Coulter Life Sciences) and finally quantified using Qubit dsDNA HS Assay Kit (Thermo Fisher Scientific) as 138 ng in total 20 µL. In both the NGS library preparation process(human genomic DNA panel and synthetic DNA panel), we used iTaq Universal SYBR Green Supermix(Biorad) with the fast protocol for all qPCR quantifications, Phusion® Hot Start Flex DNA Polymerase(NEB) with standard protocol for all PCR amplifications, DEPC water as dilution/elution buffer for all PCR mixture and elution, and SPRISelect Beads for most purification/size selection if not otherwise specified.

*Capture probe pool preparation*. Forty-two thousand 80-nt long targeting regions were selected out of the whole genome with high specificity so that there was no pseudogene impacting sequencing depth. Each target was attached with a 30-nt universal forward primer domain and a 30-nt universal reverse domain on two ends respectively for future PCR amplification. We used 2 sets of universal domains for all 42,000 targets, 21,000 targets share one set of primers. Capture probe stock containing 42,000 140-nt long sequence species was synthesized by Twist Biosciences. As shown in Fig. S1, we started with amplifying 10,000× dilution of capture probe stock for 33 cycles using Biotinylated forward primers(containing dU) and phosphorylated reverse primers, where separate tubes were used for each pair of primers. Purified amplicons of these tubes were mixed for Lamda Exonuclease (NEB) digestion of the phosphorylated reverse strands. The final products, which were single-stranded capture probes, were purified using Zymo Oligo Clean and Concentrator and quantified using Qubit™ ssDNA Assay Kits.

*Hybrid-capture and library preparation*. Fifteen microliters of the target pool with indexed genomic DNA fragments were first treated with adapter blocker sequences (to prevent long adapter sequences from forming daisy chains) and purified.

Instead of using water as an elution buffer, we used 19 µL of the mixture of capture probe pool and IDT Hybridization Buffer (IDT) to elute the mixture of target and capture probe mix for hybridization capture. The average final concentration for the capture probe mix was 30 pM per capture probe species. The mixture was denatured at 95 °C for 5 min to dissociate all double-stranded target oligos and then incubated at 65 °C for at least 16 h in order to achieve high target capture efficiency. After hybridization, the hybridization mixture was transferred to the tube containing Beads mixture (4 µL Dynabeads MyOne Streptavidin T1 pre-incubated and washed with ERCC oligos mix for surface coating). Then the tube content was fully mixed and incubated at 65 °C for an additional 45 min, while vortexed every 10 min for beads to remain in suspension. Subsequently, in order to remove un-captured genomic DNA fragments, the bead solution was washed using 100 µL preheated washing buffer under 65 °C with 5 min incubation. Then the solution was washed an additional 3 times using room temperature washing buffer with 2 min incubation at 65 °C. Post-wash product was transferred to a new tube and treated with USER enzyme and TE buffer to release bound DNA. A solution containing released DNA was qPCR quantified using index primers. Additional amplification using index primer was performed if the quantity was too low for NGS sequencing. When additional PCR was performed, the purified post-PCR product was quantified again using Qubit dsDNA HS Assay Kit (Thermo Fisher Scientific).

*Data processing*. The fasta file was aligned to reference sequences using Bowtie2 alignment software[33]. We found that 39,145 out of 42,000 probes captured target withs high specificity, and 38,040 out of the 39,145 probes had reads, which could be used in the DLM as $\log_{10}(Depth)$.

**Synthetic DNA NGS library preparation**. Synthetic DNA NGS experiment had different target and capture probe preparation process from using human genomic DNA. As shown in Fig. S2, 7373 110-nt target sequences were selected out of 100,000-plex synthetic panel synthesized by Twist Bioscience and attached with 20-nt universal regions at both ends of the targets during synthesis. Similar to using a human panel, this stock solution was diluted and then amplified to a final concentration of 50 pM of each species, using universal forward primer and reverse primer(biotinylated&with dU), where the synthesized forward strand worked as the target strand and the biotinylated reverse strand worked as the capture probe. 20 times of Blocker mix was spiked into this amplified product to block non-specific capturing caused by universal primer region and enzymatic extension from leftover primers. The blocker mix was a mixture of primer sequences with 3′ decorations. This mixture was allowed to denature at 95 °C for 3 min and re-hybridize at 55° for 3 h. The hybridization mixture was treated similarly to the human genomic DNA panel, as was captured by Streptavidin T1 Beads and released by USER enzyme after multiple times of washing. The purified final product was diluted and attached with an index through PCR with index primers. Final quantification of the purified library was performed using Quibit and then diluted for downstream NGS sequencing. Standard BWA alignment was used for extracting alignment information from the post-run FASTA file. One alignment could pass the filter if no fewer than 90 out of the 150 bases were a perfect match.

Here, we describe the experimental methods used to perform the fluorescence characterization of hybridization kinetics. The data fitting and modeling methods of strand displacement are described in later sections.

### Fluorescence studies

*Oligonucleotide synthesis and formulation*. All DNA oligonucleotides were ordered from Integrated DNA Technologies in 100 µM LabReady format, pre-suspended in Tris EDTA buffer. Target T, Complementary probe C, and protector P oligonucleotides were ordered as standard desalted oligos at the 25 nanomole scale. Fluorophore F and quencher Q oligonucleotides were ordered as HPLC purified oligos at the 250 nanomole scale. All oligonucleotide stock solutions were quantitated by Nanodrop to determine concentration.

Working secondary stocks of the oligonucleotides were prepared at the following concentrations: 5 µM for C, 10 µM for T, 10 µM for P, 5 µM for F, and 25 µM for Q. Stocks of QT for Hybridization were prepared for each target T by mixing 10 µL Q, 15 µL T, and 75 µL 5× PBS. Similarly, hybridization stocks of FP were prepared for each probe by mixing 10 µL F, 15 µL C, and 75 µL 5× PBS. Stocks of QPFC complex were prepared for each target by mixing 10 µL Q, 15 µL P, 10 µL F, 15 µL C, and 50 µL 5× PBS.

These secondary stocks were then thermally annealed, cooling from 95 °C to 20 °C over the course of 75 min. Unless otherwise specified, all the annealing processes mentioned below occurred over 75 min, cooling from 95 °C to 20 °C.

*Fluorescence observation of hybridization and strand displacement kinetics*. Fluorescence experiments were performed using two Horiba Fluoromax-4 instruments, and a 4-sample changer. The slit sizes used for the excitation and emission monochromators were 8 nm and 8 nm. For each 50 s time point, each cuvette's fluorescence was measured for 9 s.

For each hybridization experiment, an appropriate amount of FP stock was pipetted into each cuvette at the beginning of the experiment, and the fluorescence was allowed to stabilize over the course of 5 to 30 min (fluorescence was observed during this time). Subsequently, the cuvettes were removed from the instruments,

and an appropriate amount of the QT stock was added to the cuvette, following which the cuvette was placed back into the instrument.

For each strand displacement experiment, an appropriate amount of the QPFC solution was pipetted into each cuvette at the beginning of the experiment, and the fluorescence was allowed to stabilize over the course of 5 to 30 min (fluorescence was observed during this time). Subsequently, the cuvettes were removed from the instruments, and an appropriate amount of the target T was added to the cuvette, following which the cuvette was placed back into the instrument.

As shown in Fig. S3a, for each hybridization experiment, positive and negative control experiments were performed to allow mathematical conversion of observed fluorescence values into instantaneous hybridization yields. Negative control experiments included only the FP species, and show the high fluorescence corresponding to 0% yield. Positive control experiments included the FP and QT species thermally annealed (at the hybridization experiment concentrations), and show the low fluorescence corresponding to 100% yield.

As shown in Fig. S3b, for each strand displacement experiment, positive and negative control experiments were performed to allow mathematical conversion of observed fluorescence values into instantaneous reaction yields. Negative control experiments included only the QPFC species, and show the low fluorescence corresponding to 0% yield. Positive control experiments included the QPFC and T species thermally annealed (at the strand displacement experiment concentrations), and show the high fluorescence corresponding to 100% yield.

All control experiments were performed under the same reaction temperature, using the same machine and same cuvette position, in order to avoid system inconsistency.

The yield calculation of each experimental data point can be described as the following equation:

$$\text{Yield}_n = \frac{Fluo_n - Fluo_{\text{Negative Ctrl}}}{Fluo_{\text{Positive Ctrl}} - Fluo_{\text{Negative Ctrl}}} \tag{1}$$

**Reporting summary**. Further information on research design is available in the Nature Research Reporting Summary linked to this article.

## Data availability

The sequences of the DNA oligos used for the manuscript, the read depth and GC content of NGS probes, the oligo concentrations used for fluorescence experiments, the best-fit rate constants, and the values of the manually-constructed features for the WNV model are included in Supplementary Data 1–6. The original raw fluorescence data from our single-plex kinetics experiments and the raw NGS data for measuring read depth can be found at https://figshare.com/articles/dataset/A_Deep_Learning_Model_for_Predicting_Next-Generation_Sequencing_Depth_from_DNA_Sequence/14462103.

## Code availability

We provide our DLM software code and installation/usage instructions at https://github.com/XiangjiangWang/A-Deep-Learning-Model-for-Predicting-Next-Generation-Sequencing-Depth-from-DNA-Sequence.

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

## Acknowledgements

This work was funded by NIH grant R01HG008752 and U01CA233364 to D.Y.Z. The authors thank Jianyi Nie for their editorial assistance.

## Author contributions

J.X.Z., B.Y., A.G., M.X.W., A.P., and D.Y.Z. conceived the project. P.D., J.X.Z., M.X.W., Y.J.C., K.Z., and J.Z.F. performed the experiments. N.D. performed hybridization reaction model fitting and selection. A.G., B.Y., and M.X.W. created and optimized the D.L.M. J.L. processed sequencing data. A.P. and D.Y.Z. wrote the manuscript with input from all authors.

## Competing interests

There is a patent pending on X-probes used in this work. There is a patent-pending on the WNV model of rate constant prediction. M.X.W. declares a competing interest in the form of consulting for Nuprobe. D.Y.Z. declares a competing interest in the form of consulting for and equity ownership in Nuprobe, Torus Biosystems, and Pana Bio. The remaining authors declare no competing interests.
