## [Peer Review File · Nature Communications]

REVIEWER COMMENTS

Reviewer #1 (Remarks to the Author):

Zhang et al present a new method where they use a deep neural network model to predict reading depth. They use an approach based on recurrent neural networks. The article is well written and does present an interesting contribution to the field of using ML in biological sequence context. I think it is suitable for publication in Nature Communications, as long as authors address the issues raised below:

1) In my understanding, when the authors validated the model, they used all data in the cross-validation, by using 95% for training and 5% for validation, is that correct? Ideally, they should leave out a fraction of the original data completely out of the model, not even use it in the cross-validation, and then report the model performance already left out. Otherwise, I am concerned it might bias them for example in choosing the model hyperparameters to optimize for their dataset. I would be interested to see how the model performs on synthetic panel when only trained on the SNP panel (and vice versa). In a similar fashion, how does the model perform on predicting strand displacement kinetics if it is only trained on the NGS dataset, and how does it perform on the NGS dataset if it is only trained on the strand displacement kinetics?

2) Following up on the comparison with strand displacement, the authors hypothesize the physical effect that causes different NGS depths is the k_{on} of the probe hybridization. It would be helpful to explore in more detail this hypothesis.

Authors have previously published a model (Predicting DNA hybridization kinetics from sequence, Zhang et al, Nat Chem 2017) where they used a weighted feature model to predict hybridization kinetics. How does the k_{on} prediction based on the weighted kinetics model correlate with the NGS depth prediction? And, how does the network trained on NGS perform on the hybridization dataset from the Nature Chem paper? Does its prediction correlate with k_{on} ?

3) In Figure 2d, there are a couple of outliers that have much lower observed depth than predicted. Have the authors checked if these sequences have some particular features in common (GC content, length, etc?)

4) What was the length distribution of the sequences used in this work? Does the model perform differently for short vs long sequences in terms of accuracy of prediction?

Reviewer #2 (Remarks to the Author):

Zhang et al have proposed a Deep Learning Model for Predicting NGS Sequencing Depth from DNA Sequence. NGS based assays are now widely used in genomics in both germline and somatic areas. Multiple clinical assays have been deployed by vendors. Laboratories have also developed their own lab developed assays (LDT) using tools available from vendors. Over the last 10 years, NGS has improved significantly but problem areas such as high GC, repeat regions and complex sequence regions cannot be sequenced at a high depth. Sequence/ strand bias can also significantly impact the ability to interpret data. The authors have developed a DLM which utilizes both the sequence and the target information to predict the sequencing depth from a DNA sequence. The data produced from the two datasets is encouraging but it is not clear if it will be useful for clinical laboratories as the issues specifically described here above have not been addressed. It is important to see specific examples of different pathogenic - simple and complex variants to demonstrate the effectiveness of the DLM prediction tool when designing NGS assays. Data presented is on SNP panels. Lastly, the application of DLM to AT rich non-coding regions will be useful for the reader as more and more intronic pathogenic variants are being identified by Whole Genome Sequencing and RNA sequencing.

Reviewer #3 (Remarks to the Author):

In this paper, Zhang et al. proposed a deep learning model for predicting NGS sequencing depth. This problem is very important as empirical optimization is time consuming. The writing of this paper should be improved. Most of the text in the results section should belong to methods section.

Comments.

1. Most of the text in the Design of the Deep Learning Model section should be put in the methods section. This paper right now has no methods section, which makes it very difficult to follow.

2. Fig 2a and Fig 2b are not particularly interesting since they are common observations in most machine learning models.

3. Fig 2b is mentioned after Fig 2e. Authors should reorder figure panels.

4. No comparison approaches. The only comparison approach is an expert system based on weighted neighbor voting and the ablation studies in fig 5. However, since the proposed model is a deep learning model, it is important to compare it with other classic ML models. It is also important to know if any component of the proposed DL model is not helpful and can be excluded for simplicity.

5. Potential overfitting of the deep learning model. Is it possible to perform cross-dataset validation? Train on NGS depth human and test on NGS depth synthetic?

6. Running time of the proposed DL model is missing. Is the training/test time of such model time-consuming?

Reviewer #1 (Remarks to the Author):

Zhang et al present a new method where they use a deep neural network model to predict reading depth. They use an approach based on recurrent neural networks. The article is well written and does present an interesting contribution to the field of using ML in biological sequence context. I think it is suitable for publication in Nature Communications, as long as authors address the issues raised below:

1) In my understanding, when the authors validated the model, they used all data in the cross-validation, by using 95% for training and 5% for validation, is that correct? Ideally, they should leave out a fraction of the original data completely out of the model, not even use it in the cross-validation, and then report the model performance already left out. Otherwise, I am concerned it might bias them for example in choosing the model hyperparameters to optimize for their dataset.

Yes, this is correct. We did 20-fold cross validation and in each fold 95% of the entire dataset was used for training and 5% for validation. We thank the reviewer for their recommendation to use an independent test set. To address this, we have designed and experimentally tested a new panel using the same library preparation method as the SNP panel. We first trained the model on the SNP dataset and then used it to predict the read depth of the new panel. The model predicted read depth to within a factor of 3 with 89% accuracy on the new panel (Fig. 2d). Therefore, we believe that the model is generalizable to completely new datasets.

We have updated the main text as follows (starting at page 5 bottom right):

“To validate our DLM in the practical scenario of optimizing new panels with a model trained on existing panels, we designed and tested the lncRNA panel comprising 2,000 DNA probes synthesized by Twist Biosciences. The lncRNA panel has the same library preparation method (experimental workflow, probe length, hybridization temperature, sample type, etc.) as the SNP panel, but differs in probe sequences, experimental operator, donor of DNA sample, sequencing instrument and batch of reagents. Fig. 2d shows the predictions of lncRNA panel produced by a DLM trained on SNP panel with early-stop at epoch 250. Read depth of the lncRNA panel is scaled so that the average read depth is the same as the SNP panel. Despite the RMSE, F2err and F3err of lncRNA panel being slightly worse than the SNP panel (0.326 vs. 0.301, 30.4% vs. 20.9% and 11.04% vs. 7.31%), the performance decrease may be attributed to experimental variations that are not related to the library preparation method. It is important to point out that the DLM was trained on read depth measured with only one NGS library of SNP panel, which greatly reduced the cost of training such a model. The results from lncRNA panel indicate that the DLM is capable of generalizing different panels with the same library preparation method while being robust against experimental variations.”

I would be interested to see how the model performs on synthetic panel when only trained on the SNP panel (and vice versa).

We have performed this analysis as recommended by the reviewer and the results can be found in Supplementary Section S3 (starting at page 35 bottom). In short, we saw a weak correlation between the predicted and observed read depth when the model was trained on either the SNP panel or the synthetic panel and tested on the other one. The correlation was higher when the model was trained on the SNP panel and tested on the lncRNA panel since

these two panels used the same library preparation method. Below is the quoted text from the supplementary:

“Note that if a probe P is fed to a DLM trained on a NGS panel A, the prediction is the expected read depth of probe P in the context of panel A, which does not have to correlate with the observed read depth of probe P in panel B. Such correlation depends on the library preparation methods of panel A and B. For the DLM trained on the SNP panel, the Pearson correlation coefficient between the predicted and observed $\log_{10}(\text{NormDepth})$ of the lncRNA panel is 0.728, while the Pearson correlation coefficient of the synthetic panel is only 0.319. This is because the SNP panel and the lncRNA panel use the same library preparation method. We noticed that the predicted read depth of probes in the synthetic panel is above average (zero $\log_{10}(\text{NormDepth})$ correspond to average read depth), which might be because those probes are specially designed to have high hybridization yield instead of chosen from human genome.”

In a similar fashion, how does the model perform on predicting strand displacement kinetics if it is only trained on the NGS dataset, and how does it perform on the NGS dataset if it is only trained on the strand displacement kinetics?

To address these questions, we have compared the predicted read depth and the observed rate constant for the strand displacement dataset, using models trained on the two NGS panels. We have also compared the predicted strand displacement rate constant and the observed read depth of the two NGS panels, using the model trained simultaneously on both the hybridization and strand displacement datasets. However, we did not see a significant correlation in either case.

The results can be found in Supplementary Section S3 (starting at page 35 bottom), which we have updated with the following text:

“Although the true values of hybridization rate constants for our NGS probes are not known, we could predict the rate constants of those probes with the DLM trained simultaneously on our hybridization (HYB) and strand displacement (DSP) dataset. On the contrary, we could predict the read depth of the probes in the HYB or DSP dataset with a DLM trained on one of the NGS dataset. However, we did not see a significant correlation in either case. There should be two possible explanations, 1) the wide difference of probe length between NGS panels and kinetics experiments makes the predictions of the hybridization rate constant of NGS probes inaccurate, 2) there is only a weak correlation between the NGS read depth and the hybridization rate constant of a certain probe. Note that the features of the DSP dataset are calculated not only based on the sequences of the probes, but also the sequences of the protectors that greatly reduce open base probabilities, resulting in low predicted read depth of the DSP dataset.”

2) Following up on the comparison with strand displacement, the authors hypothesize the physical effect that causes different NGS depths is the k_{on} of the probe hybridization. It would be helpful to explore in more detail this hypothesis.

To address this comment, we have predicted the k_{on} of the probes in both NGS panels using the model trained simultaneously on both the hybridization and strand displacement datasets and we did not see a correlation between the predicted k_{on} and the observed read depth. A possible explanation is that the wide difference in probe length between the kinetics dataset (36nt) and the two NGS datasets (80nt and 110nt) makes the predicted k_{on} inaccurate.

The results can also be found in Supplementary Section S3 (starting at page 35 bottom).

Authors have previously published a model (Predicting DNA hybridization kinetics from sequence, Zhang et al, Nat Chem 2017) where they used a weighted feature model to predict hybridization kinetics. How does the k_{on} prediction based on the weighted kinetics model correlate with the NGS depth prediction?

And, how does the network trained on NGS perform on the hybridization dataset from the Nature Chem paper? Does its prediction correlated with k_{on} ?

We mentioned in the main text that the experimental hybridization rate constants used in this paper are taken from the previous Nat Chem 2017 paper (page 7 middle left). Fig. 4e shows that the predictions of the weighted neighbor model and the DLM are very similar, so we did not try the weighted neighbor model on the NGS datasets. We did use the models trained on the NGS datasets to predict the read depth of the probes in the hybridization dataset. However, we did not see a significant correlation.

The results can also be found in Supplementary Section S3 (starting at page 35 bottom).

3) In Figure 2d, there are couple of outliers that have much lower observed depth than predicted. Have the authors checked if these sequences have some particular features in common (GC content, length, etc?)

Following the reviewer's suggestion, we observed that the outliers with low observed read depth tend to have lower GC content, which directly correlates with hybridization energy. The possible reason why the read depth is overestimated can be found in the main text (starting at page 4 bottom right):

“From this figure, we see that a significant contributor to our DLM's RMSE is a subset of DNA sequences that are observed to have very low $\log_{10}(\text{Depth})$ (e.g., 0.3, corresponding to a depth of 2), but predicted to have $\log_{10}(\text{Depth})$ between 1 and 3.3. Further investigating the probe sequences, we found that most of the probes have low G/C content. Our interpretation of this phenomenon is that probes with lower expected read depth (e.g. probes with low G/C content) are more sensitive to random fluctuations (probe synthesis yield, hybridization yield, binding to plasticware, bridge PCR efficiency during Illumina NGS, etc.), which would bias the observed $\log_{10}(\text{Depth})$. For example, suppose the expected read depth for a certain probe is 50 and the random fluctuation is ± 45 with uniform distribution, then the observed depth ranges from 5 to 95 while the observed $\log_{10}(\text{Depth})$ ranges from 0.70 to 1.98. Note that the expected $\log_{10}(\text{Depth})$ is 1.70; thus there is a higher probability of $\log_{10}(\text{Depth})$ being lower than expected than being higher than expected ($1.70 - 0.7 < 1.98 - 1.70$). If the DLM predicts the expected $\log_{10}(\text{Depth})$, then there would be more probes whose observed $\log_{10}(\text{Depth})$ is lower than the predicted $\log_{10}(\text{Depth})$. However, the DLM cannot explain those random fluctuations solely based on probe sequences.”

4) What was the length distribution of the sequences used in this work? Does the model perform differently for short vs long sequences in terms of accuracy of prediction?

All the probes in the same NGS panel have the same length. Probe length of the SNP panel is 80nt (page 3 bottom right) and the synthetic panel is 110nt (page 6 bottom left). Probe length in the kinetics dataset is 36nt and the length of protectors varies (page 7 middle left). We believe that a systematic training and assessment of the model for many different lengths is beyond the scope of this paper.

Reviewer #2 (Remarks to the Author):

Zhang et al have proposed a Deep Learning Model for Predicting NGS Sequencing Depth from DNA Sequence. NGS based assays are now widely used in genomics in both germline and somatic areas. Multiple clinical assays have been deployed by vendors. Laboratories have also developed their own lab developed assays (LDT) using tools available from vendors. Over the last 10 years, NGS has improved significantly but problems areas such as high GC, repeat regions and complex sequence region cannot be sequenced at a high depth. Sequence/ strand bias can also significantly impact the ability interpret data. The authors have developed a DLM which utilizes both the sequence and the target information to predict the sequencing depth from a DNA sequence. The data produced from the two datasets is encouraging but it is not clear if it will be useful for clinical laboratories as the issues specifically described here above have not been addressed. It is important to see specific examples of different pathogenic - simple and complex variants to demonstrate the effectiveness of the DLM prediction tool when designing NGS assays. Data presented is on SNP panels. Lastly, the application of DLM to AT rich non- coding regions will be useful for the reader as more and more intronic pathogenic variants are being identified by Whole Genome Sequencing and RNA sequencing.

To further show the applicability of our prediction algorithms, we have added new experimental NGS data on a human long non-coding RNA (lncRNA) panel. Our lncRNA panel includes some probes with low AT fractions down to 20%, to address the reviewer's concern. The observed read depth, predicted read depth and GC content of the lncRNA probes can be found in the Excel file accompanying the manuscript. We have also updated the main text as follows (starting at page 5 bottom right):

“To validate our DLM in the practical scenario of optimizing new panels with a model trained on existing panels, we designed and tested the lncRNA panel comprising 2,000 DNA probes synthesized by Twist Biosciences. The lncRNA panel has the same library preparation method (experimental workflow, probe length, hybridization temperature, sample type, etc.) as the SNP panel, but differs in probe sequences, experimental operator, donor of DNA sample, sequencing instrument and batch of reagents. Fig. 2d shows the predictions of lncRNA panel produced by a DLM trained on SNP panel with early-stop at epoch 250. Read depth of the lncRNA panel is scaled so that the average read depth is the same as the SNP panel. Despite the RMSE, F2err and F3err of lncRNA panel being slightly worse than the SNP panel (0.326 vs. 0.301, 30.4% vs. 20.9% and 11.04% vs. 7.31%), the performance decrease may be attributed to experimental variations that are not related to the library preparation method. It is important to point out that the DLM was trained on read depth measured with only one NGS library of SNP panel, which greatly reduced the cost of training such a model. The results from lncRNA panel indicate that the DLM is capable of generalizing different panels with the same library preparation method while being robust against experimental variations.”

Complex variants such as microsatellites and telomeres are scientifically interesting but likely beyond the scope of an initial study on prediction of NGS depth from DNA sequence, partially because we currently lack even good experimental methods for accurately aligning/mapping these sequences. As with all supervised machine learning approaches, we require high quality labeled data for training of the machine learning model.

Reviewer #3 (Remarks to the Author):

In this paper, Zhang et al. proposed a deep learning model for predicting NGS sequencing depth. This problem is very important as empirical optimization is time consuming. The writing of this paper should be improved. Most of the text in the results section should belong to methods section.

Comments.

1. Most of the text in the Design of the Deep Learning Model section should be put in the methods section. This paper right now has no methods section, which makes it very difficult to follow.

We created a new Methods section including the design of the DLM and the method of training and validation of the DLM.

2. Fig 2a and Fig 2b are not particularly interesting since they are common observations in most machine learning models.

Fig. 2a would be helpful for readers not familiar with machine learning and the NGS data analysis pipeline, and Fig. 2b compares the GC content distribution of the NGS panels, which clarifies the differences between the three panels. To be on the safe side we have kept these figures in the main text, but we are happy to move them to the supplement if required.

3. Fig 2b is mentioned after Fig 2e. Authors should reorder figure panels.

We moved Fig. 2b to the last.

4. No comparison approaches. The only comparison approach is an expert system based on weighted neighbor voting and the ablation studies in fig 5. However, since the proposed model is a deep learning model, it is important to compare it with other classic ML models. It is also important to know if any component of the proposed DL model is not helpful and can be excluded for simplicity.

We have compared the DLMs with a linear regression model and the results can be found in Fig. 2b and Supplementary Section S3 (page 35 top). The DLMs outperformed linear regression models in all cases.

In the first submission, we have evaluated different features of the DLM by removing each single feature and compared the prediction accuracies (Fig. 5). We concluded that 3 of the global features can be removed with only a small decrease in accuracy.

5. Potential overfitting of the deep learning model. Is it possible to perform cross-dataset validation? Train on NGS depth human and test on NGS depth synthetic?

In the first submission, we applied early-stop based on the loss of validation set in each epoch (Fig. 3a) to prevent overfitting. We further validated the model by designing and experimentally testing a new human panel and predicted the read depth with the model trained on the SNP panel. The model predicted read depth to within a factor of 3 with 89% accuracy on the new panel. Therefore, we believe that we are not overfitting the model and it is generalizable to

completely new datasets. For the new panel, we have updated the main text as follows (starting at page 5 bottom right):

“To validate our DLM in the practical scenario of optimizing new panels with a model trained on existing panels, we designed and tested the lncRNA panel comprising 2,000 DNA probes synthesized by Twist Biosciences. The lncRNA panel has the same library preparation method (experimental workflow, probe length, hybridization temperature, sample type, etc.) as the SNP panel, but differs in probe sequences, experimental operator, donor of DNA sample, sequencing instrument and batch of reagents. Fig. 2d shows the predictions of lncRNA panel produced by a DLM trained on SNP panel with early-stop at epoch 250. Read depth of the lncRNA panel is scaled so that the average read depth is the same as the SNP panel. Despite the RMSE, F2err and F3err of lncRNA panel being slightly worse than the SNP panel (0.326 vs. 0.301, 30.4% vs. 20.9% and 11.04% vs. 7.31%), the performance decrease may be attributed to experimental variations that are not related to the library preparation method. It is important to point out that the DLM was trained on read depth measured with only one NGS library of SNP panel, which greatly reduced the cost of training such a model. The results from lncRNA panel indicate that the DLM is capable of generalizing different panels with the same library preparation method while being robust against experimental variations.”

We performed cross-dataset validation and the results can be found in Supplementary Section S3 (page 35 bottom):

“Note that if a probe P is fed to a DLM trained on a NGS panel A, the prediction is the expected read depth of probe P in the context of panel A, which does not have to correlate with the observed read depth of probe P in panel B. Such correlation depends on the library preparation methods of panel A and B. For the DLM trained on the SNP panel, the Pearson correlation coefficient between the predicted and observed $\log_{10}(\text{NormDepth})$ of the lncRNA panel is 0.728, while the Pearson correlation coefficient of the synthetic panel is only 0.319. This is because the SNP panel and the lncRNA panel use the same library preparation method. We noticed that the predicted read depth of probes in the synthetic panel is above average (zero $\log_{10}(\text{NormDepth})$ correspond to average read depth), which might be because those probes are specially designed to have high hybridization yield instead of chosen from human genome.”

6. Running time of the proposed DL model is missing. Is the training/test time of such model time-consuming?

For the SNP panel (38,040 probes), training stops at epoch 250 and the training time for each epoch is roughly 10 seconds while taking less than 3 gigabytes memory of a graphics processing unit (batch size of 999), and feature generation using Nupack takes about 0.5 second per probe sequence on a conventional desktop computer. This is also mentioned at the end of Methods section (page 3).

REVIEWER COMMENTS

Reviewer #1 (Remarks to the Author):

The authors have addressed my comments to a satisfactory degree. While I would have wished for them to more carefully explore the accuracy of their k_{on} prediction and the sensitivity of the model to the probe lengths, authors consider this to be beyond the scope of their paper, and I tend to agree.

Reviewer #2 (Remarks to the Author):

The revised manuscript has addressed the questions raised by reviewers in a step by step manner. The authors have also added more data to show the efficiency of their DLM model to predict NGS sequencing depth. Though some questions such as complex molecular events as mentioned by the authors are not within the scope of their manuscript. Taking that into consideration the current trend on of moving towards whole genome sequencing it will be useful for the reader to get side by side comparison of a

1. Full exome modeled using the DLM to predict sequence depth
2. or a chromosome with high GC rich sequences modeled
3. Take a panel such as Autism which has a large number of genes and model it.

This is necessary for several reasons-

1. WGS affords more uniformity in sequencing at a lower depth than exome / panel sequencing . If the DLM can facilitate better prediction of the design it will be very useful for clinical labs
2. Not all labs can switch to genome sequencing due to cost
3. a small portion of the genome cannot be sequenced.

Reviewer #3 (Remarks to the Author):

The authors have addressed all of my questions.

Reviewer #1 (Remarks to the Author):

The authors have addressed my comments to a satisfactory degree. While I would have wished for them to more carefully explore the accuracy of their k_{on} prediction and the sensitivity of the model to the probe lengths, authors consider this to be beyond the scope of their paper, and I tend to agree.

We thank the reviewer for all the comments. We will further explore the relationship between reads depth, k_{on} and probe length in future studies.

Reviewer #2 (Remarks to the Author):

The revised manuscript has addressed the questions raised by reviewers in a step by step manner. The authors have also added more data to show the efficiency of their DLM model to predict NGS sequencing depth. Though some questions such as complex molecular events as mentioned by the authors are not within the scope of their manuscript. Taking that into consideration the current trend on of moving towards whole genome sequencing it will be useful for the reader to get side by side comparison of a

- 1. Full exome modeled using the DLM to predict sequence depth*
- 2. or a chromosome with high GC rich sequences modeled*
- 3. Take a panel such as Autism which has a large number of genes and model it.*

This is necessary for several reasons-

- 1. WGS affords more uniformity in sequencing at a lower depth than exome / panel sequencing . If the DLM can facilitate better prediction of the design it will be very useful for clinical labs*
- 2. Not all labs can switch to genome sequencing due to cost*
- 3. a small portion of the genome cannot be sequenced.*

We thank the reviewer's suggestion of applying the DLM to existing panels. Two commercial panels were modeled with the DLM trained on the SNP panel: xGen Exome Research Panel v2 and xGen Acute Myeloid Leukemia Cancer Panel. The former covered human whole exome and the latter targeted more than 260 genes. We chose these two panels since they had public probe sequences and library preparation methods. The results can be found in Supplementary Section S3 (starting at page 36 bottom), which we have updated with the following text and figure:

"We further applied the DLM trained on the SNP panel to two commercial NGS panels with public probe sequences: xGen Exome Research Panel v2 (Integrated DNA Technologies), abbreviated as the exome panel, and xGen Acute Myeloid Leukemia Cancer Panel (Integrated DNA Technologies), abbreviated as the AML panel. These two panels have the same library preparation method (120nt probe length and 65°C hybridization temperature) but different probe sequences: the exome panel has 415,115 probes covering human whole exome and the AML panel has 11,731 probes targeting more than 260 human genes. The prediction results are shown in Fig. S35. The exome panel and the AML panel have higher median predicted read depth than the SNP panel and the lncRNA panel, which might be attributed

to longer probe length (120nt vs. 80nt). The synthetic panel has the highest median predicted read depth and the lowest variation since its probe sequences are artificially designed for hybridization.”

FIG. S35: The DLM trained on the SNP panel is applied to commercial panels with public probe sequences. SNP: human single nucleotide polymorphisms panel, 39,145 probes. lncRNA: human long non-coding panel, 2000 probes. synthetic: artificially designed synthetic sequences for information storage, 7,373 probes. exome: xGen Exome Research Panel v2, 415,115 probes. AML: xGen Acute Myeloid Leukemia Cancer Panel, 11,731 probes. Color code represents library preparation method.

Reviewer #3 (Remarks to the Author):

The authors have addressed all of my questions.

We thank the reviewer for all the comments.

REVIEWERS' COMMENTS

Reviewer #2 (Remarks to the Author):

The authors have addressed my comments.

Reviewer #2 (Remarks to the Author):

The authors have addressed my comments.

We thank the reviewer for all the comments.